# Facilitating Resilience during an African Swine Fever Outbreak in the Austrian Pork Supply Chain through Hybrid Simulation Modelling

Yvonne Kummer [1,*], Christian Fikar [2,†], Johanna Burtscher [3], Martina Strobl [4], Reinhard Fuchs [5,6], Konrad J. Domig [3] and Patrick Hirsch [1]

1   Institute of Production and Logistics, Department of Economics and Social Sciences, University of Natural Resources and Life Sciences, Feistmantelstraße 4, 1180 Vienna, Austria; patrick.hirsch@boku.ac.at
2   Faculty VII for Life Sciences: Food, Nutrition and Health, University of Bayreuth, Fritz-Hornschuch-Strasse 13, 95326 Kulmbach, Germany; christian.fikar@uni-bayreuth.de
3   Institute of Food Science, Department of Food Science and Technology, University of Natural Resources and Life Sciences, Muthgasse 18, 1190 Vienna, Austria; johanna.burtscher@boku.ac.at (J.B.); konrad.domig@boku.ac.at (K.J.D.)
4   Austrian Competence Centre for Feed and Food Quality, Safety and Innovation (FFoQSI GmbH), Technopark 1C, 3430 Tulln, Austria; martina.strobl@boku.ac.at
5   Department for Data, Statistics and Risk Assessment, Austrian Agency for Health and Food Safety (AGES), Zinzendorfgasse 27, 8010 Graz, Austria; reinhard.fuchs@ages.at
6   Institute of Systems Sciences, Innovation and Sustainability Research, University of Graz, Merangasse 18, 8010 Graz, Austria
*   Correspondence: yvonne.kummer@boku.ac.at; Tel.: +43-1-47654-73418
†   Chair of Food Supply Chain Management.

**Abstract:** This study aimed to simulate the impact of an African swine fever (ASF) outbreak in Austria. ASF is one of the most significant and critical diseases for the global domestic pig population. Hence, the authors evaluated control strategies and identified bottlenecks during an ASF outbreak. A hybrid approach was selected, including discrete-event and agent-based simulation. An extended Susceptible-Exposed-Infectious-Recovered (SEIR) model (within a pig farm) and a standard SEIR model (between pig farms) were used to simulate the chain of infection. A total of 576 scenarios with several parameter variations were calculated to identify the influence of external factors on key performance indicators. The main results show a comparison between two control strategies anchored in law: a standard strategy (SS) and a preventive culling strategy (SC). The calculated scenarios show a difference between these strategies and indicate that with SC during an outbreak, fewer farms would be infected, and fewer pigs would be culled. Furthermore, specific geographical areas were identified, which—due to their density of pigs and farms—would be severely affected in case of an ASF outbreak. The analysis of bottlenecks in rendering plants (RPs) showed an increase in the number of days RPs were overutilized as the transmission rate increased. In addition, SS caused more days of overutilized RPs than SC.

**Keywords:** African swine fever; simulation; pork production; resilience; control strategy; decision support system

## 1. Introduction

The African swine fever (ASF) virus is one of the most important pathogens affecting the global domestic pig population due to its socio-economic impact and the complexity of preventing its spread [1–4]. Several reasons make the virus hazardous: (i) its multiple modes of transmission and the role of wild boars therein, (ii) the fact that fatality is nearly 100%, and (iii) the long persistence in the environment [5–7]. Hence, ASF is a notifiable disease that must be reported to the World Organization of Animal Health when suspected [8,9].

Given the long stability of the ASF virus, dead infected wild boars are the main cause of spread [10,11]. However, the history of the spread of ASF, since it was first discovered in Kenya in 1910 [12], shows that there have also been repeated widespread jumps of the disease [13,14]. An underlying reason is that the virus remains active in meat products and may spread through these transnationally [15]. There are various mechanisms of spread at the national level in addition to wild boars; surfaces, feed, and water can be contaminated with the ASF virus and contribute to its dissemination [9,16]. Although ASF is an epizootic, which means that the virus is not transmissible to humans [17], the economic impact of a national ASF outbreak is enormous. This is due to export bans in addition to the predicted long duration of an outbreak and the threat to food security due to restricted national trade and shrinking pig herds [18–22].

Since the publication of the Food and Agriculture Organization of the United Nations (FAO) world food security declaration in 1996, the definition of food security and the topic itself have been recognized and acknowledged globally [23]. The 1996 FAO definition states that food security exists "when all people, at all times, have physical and economic access to sufficient, safe and nutritious food to meet their dietary needs and food preferences for an active and healthy life" [24]. In 2001 the FAO expanded this definition to include not only physical and economic access but also the social aspect [25]. This definition consists of all four dimensions of food security, which according to Jones et al. [23], are availability, access, utilization and stability over time. The entire food system must be evaluated to achieve or maintain food security, which includes four areas: food production, food processing, food distribution, and food access. Hecht et al. [26] have pointed out that, in contrast to resilience research in other fields, such as agriculture and infrastructure, research on the resilience of food systems is still in its early stages. Holling [27] significantly shaped the definition of resilience by distinguishing it from the concept of stability by identifying the persistence of systems and the ability to absorb disturbances but still continue to exist as essential characteristics of resilience. Garnier [28] has defined resilience in more detail as the ability of a social system to sustain the vital structures during a crisis. Other definitions have focused on the short-term capacity for resistance or adaptation, including resistance to stress, adaptive capacity, and transformational capacity [29].

The current situation of ASF in the European Union reveals that 13,193 and 1920 cases of wild boar and domestic pig, respectively, were reported in the Animal Diseases Information System from 1 January 2021 to 30 January 2022 [30]. Austria is not affected until now (as of 4 February 2022), but ASF outbreaks in neighbouring countries have aroused national attention concerning this disease and the need for disease control strategies. This work presents a hybrid simulation (HS) model of the Austrian pork supply chain based on real data. The model contributes to an enhancement of supply chain resilience by simulating hypothetical outbreak scenarios of ASF, evaluating control strategies, and deriving recommendations for policy action. The control strategies, regulated by law, contain actions in case of confirmation or suspicion of ASF in pig holdings, slaughterhouses, or means of transport and define regulations regarding the transport of animals, contact holdings, surveillance- and protection areas, and epidemiological investigations. These regulations can be divided into two control strategies: (i) standard strategy (SS), in which all measures are taken according to the rules of the regulation, and (ii) the strategy (preventive) culling (SC), in which the legal possibilities are used to cull animals that are merely suspected of being infected with ASF when this is sufficiently justified. Furthermore, external factors, which are: (i) the location of the epicentre, (ii) the intensity of the infection event, and (iii) the type of the first infected farm, are considered. Both internal (control strategies) and external factors were evaluated together in several numerical experiments to answer the following research question: What is the impact of the two control strategies in terms of key performance indicators (KPIs) under different external factors? The considered KPIs are: (i) the number of infected farms, (ii) the number of emergency slaughtered pigs, and (iii) the number of days on which rendering plants (RPs) are overutilized. Therefore, the results can serve as a basis for concrete recommendations for relevant stakeholders

(e.g.,government agencies) in the preparedness phase. Furthermore, stakeholders are provided with a comprehensive simulation environment for training purposes.

## 2. State of the Art

Since the ASF virus has been officially known for more than 100 years, numerous publications have been published on this subject. Table 1 gives an overview of previously developed models that include simulation studies to estimate the ASF spread or other veterinary diseases and their epidemiological parameters.

Hayes et al. [51] have provided a comprehensive, up-to-date review of previous models that simulate ASF dynamics. A variety of models (see, e.g., [52–55]) are engaged in modelling or calculating different transmission pathways and transmission probabilities of ASF via experiments or simulations. Five of 34 modelling studies on ASF [48,56–59] primarily targeted the consequences of a hypothetical outbreak [51]. Only two [60,61] of the 34 included studies modelled a transmission between domestic and wild boars [51]. Hayes et al. [51] have defined four research gaps related to the limitations of previous models: (i) poor evaluation of control strategies, (ii) lack of linkage between domestic pigs and wild boars, (iii) absence of a consideration of epidemiological parameters that vary at broad scales, and (iv) lack of ensemble models. Hayes et al. [51] also addressed the fact that there are national differences in production and husbandry, which is why national studies are essential. The methodology presented in the next section partially addresses these research gaps by considering the combination of domestic pigs and wild boars and by evaluating different control strategies. Rapid adaptation to new research findings and the testing of different parameters is feasible, e.g., the possibility to vary epidemiological parameters easily. Due to the model's flexibility, it can offer results for different scenarios, thus reducing the need to combine several models to obtain a comprehensive analysis.

In contrast to earlier work, the model presented in this paper simulates a hypothetical outbreak of ASF in domestic pigs in Austria using a HS technique including transmission from wild boars. It is the first model on a national level that performs such calculations. Compared to the international literature previously presented, the authors are not aware of any studies that have used the same methodology. Additionally, the model is based on real data of the Austrian domestic pig population and thus simulates the supply chain precisely with original geographical locations of the holdings and, therefore, has the potential to be used as a Decision Support System (DSS) for the simulation of different outbreak scenarios and the evaluation of control strategies. Thus, the introduced model contributes to resilience enhancement and efficient crisis management.

**Table 1.** Overview of existing models to simulate animal disease spread in different countries.

| Model | Method | Source | Country | Disease |
|---|---|---|---|---|
| EpiMAN | The model combines a database management system (DBMS), a geographic information system (GIS), expert system elements, various models on specific aspects of foot and mouth disease (FMD) epidemiology (InterSpread), and a statistical analysis capability. | Sanson [31] | New Zealand | FMD |
| InterSpread® | Inter-farm spread model using a spatial stochastic simulation operating on the actual geography of the area. | Sanson [31] | New Zealand | FMD |
| | | Jalvingh et al. [32] | New Zealand | FMD |
| | | Martínez-López et al. [33] | Spain | FMD |
| InterCSF | Spatial, temporal, and stochastic simulation model of classic swine fever (CSF), using InterSpread as the basis. | Jalvingh et al. [34] | Netherlands | CSF |
| | | Nielen et al. [35] | Netherlands | CSF |
| | | Mangen et al. [36] | Netherlands | CSF |
| InterFMD | Stochastic and spatial simulation of the spread and control of FMD. | Velthuis and Mourits [37] | Netherlands | FMD |
| InterSpread Plus® | Stochastic, individual-based, discrete time, and spatio-temporal state transition spread of infectious disease model (using InterSpread as the basis). | Boklund et al. [38] | Denmark | CSF |
| | | Nigsch [39] | Austria | CSF |
| | | Nigsch et al. [3] | European Union | ASF |
| | | Hiesel et al. [40] | Austria | FMD |
| AusSpread | Stochastic spatial simulation of the spread and control of FMD at a regional scale. | Garner and Beckett [41] | Australia | FMD |
| | | Roche et al. [42] | Australia | FMD |
| NAADSM | Spatial, stochastic, state transition simulation model. | Pendell et al. [43] | United States of America | FMD |
| | | Harvey et al. [44] | United States of America and Canada | FMD |
| | | Lee et al. [20] | Vietnam | ASF |
| Be-FAST | Discrete time stochastic susceptible-infected model (within farm); spatial stochastic individual-based model (between farms). | Martínez-López et al. [45] | Spain | CSF |
| DTU-DADS | Spatial, stochastic simulation model (between-farm spread simulated using agent-based modelling (ABM), within-farm spread modelled using a compartmental model). | Halasa et al. [46] | Denmark | FMD |
| | | Dórea et al. [47] | Sweden | FMD |
| | | Halasa et al. [48] | Denmark | ASF |
| EuFMDis | Multi-country spatially explicit simulation model with equation-based (spread within a herd) and data-driven individual-based modelling (spread between herds). | Bradhurst et al. [49] | European Union | FMD |
| | | Marschik et al. [50] | Austria | FMD |

## 3. Methodology

The developed simulation model builds on real data and uses heuristics to calculate dispersal scenarios. For this purpose, an HS approach—defined as a model that combines at least two different simulation approaches [62]—was implemented, combining discrete-event simulation (DES) and agent-based simulation (ABS). In this work, DES was used to model the business processes from primary production to the slaughterhouse (Figure 1) and ABS for the epidemiological model. To represent the epidemiological course of a disease, Susceptible-(Exposed)-Infectious-Recovered (S(E)IR) models are the most widely used method [63,64]. Other stages, such as quarantine, immunity, isolation, etc., can be included depending on the disease and the purpose of the simulation [65]. The simulation model was implemented with the software tool AnyLogic 8 University. The model uses Open Street Map data [66] and routing to locate the agents and create road connections. The routing is completed via GraphHopper in Anylogic, which calculates the shortest connections on a real road network basis. A warm-up period is needed to stabilize the delivery relationships between the single pig holdings. Due to data protection issues, the delivery relationships are computed by algorithms based on decision rules since concrete business relationships data were unavailable. The decision rules were developed based on expert interviews and literature studies and provide reliable approximations.

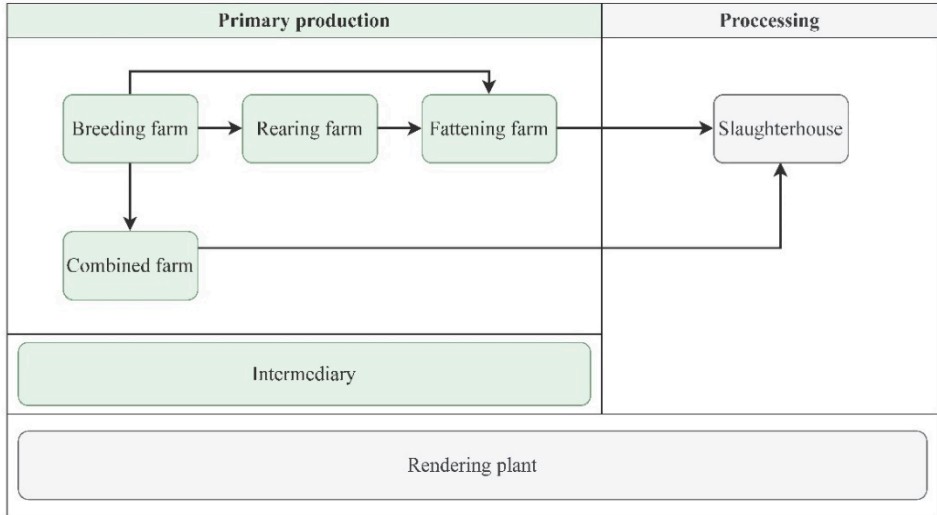

**Figure 1.** Entities in the simulation model and their relationships.

### 3.1. DES of the Pork Supply Chain

The simulated pork supply chain includes primary production and slaughtering. Other supply chain nodes are intermediaries, which act as a marketplace. These assembly centres are used within primary production to assemble homogeneous groups of pigs in the required quantity from small-scaled farms for transport to the next stage of the supply chain. The following four producing farm types are distinguished:

- Breeding farm: produces piglets for rearing or fattening purposes;
- Rearing farm: obtains piglets from breeding farms, raises them, and transport them to fattening farms;
- Fattening farm: obtains piglets from breeding or rearing farms and fattens them until slaughter;
- Combined farm: carries out pig breeding and pig fattening and occasionally obtains mother sows from breeding farms.

One additional node in the supply chain is the RP, which is used to remove contaminated animals and animal material (infected by ASF virus or other illnesses) and can be accessed from any stage of the supply chain. In our approach, the pork supply chain ends

at the slaughterhouse, where a mandatory ante- and post-mortem inspection takes place by official veterinarians. Hence, it can be assumed that infected pigs are detected at the latest here, and their meat is not processed further. Figure 1 shows all entities of the modelled supply chain.

*3.2. ABS of the ASF Outbreak*

ASF outbreaks are regulated by Regulation (EU) 2016/429 "Animal Health Law" [67] accompanied by several delegated regulations and commissions implementing regulations, such as (EU) 2021/605, and a national regulation derived from it (ASF-Regulation 2005 [68]). In addition, there are further regulations that specifically affect wild boar or regulate the movement of animals and contaminated material. The following measures in case of confirmation or suspicion of ASF in domestic pig farms are taken from the Austrian ASF-Regulation 2005 [68]:

- Culling of all pigs on an infected holding;
- Taking a sufficient number of samples and sending them to a national reference laboratory;
- Destroying all materials (e.g., waste, feeding stuff, meat) that could be contaminated;
- Carrying out epidemiological investigations;
- Establishing a protection zone of 3 km and a surveillance zone of 10 km around the infected holding immediately after confirmation. In these zones:
  a. Epidemiological investigations are carried out;
  b. All pigs are kept inside their pens;
  c. The movement and transport of pigs on public roads (with exceptions is prohibited);
- Finding contact holdings based on epidemiological investigations and applying the same measures as at an infected holding.

It is forbidden to transport domestic pigs to the slaughterhouse within 40 days in the protection zone and within 30 days in the surveillance zone after cleaning and disinfecting the last infected farm. This and the different test strategies of the zones are the two main differences relevant to the simulation.

An extended version of the Susceptible-Exposed-Infectious-Recovered (SEIR) model was implemented to simulate the spread of ASF within a pig holding. The relevant agents in this model are the pigs. The pigs per holding were assigned to several homogeneous groups and taken as an agent called "group of pigs" to reduce the number of modelled agents and save computational time. However, this does not have a significant impact on the results, as these groups can also be separated again. The extended SEIR model consists of the following steps: Susceptible-Infected-Infectious-Detected-Confirmed-Culled ($SI_1I_2DC_1C_2$). Figure 2 shows the required parameters for the $SI_1I_2DC_1C_2$ model, explained in more detail in Table 2. $\lambda$ represents the risk for an animal to become infected and is calculated as the expected number of newly infected animals $E(C)$ per time unit using Equation (1) according to Velthuis et al. [69]:

$$E(C) = S\left(1 - e^{-\beta\left(\frac{I_2}{N}\right)}\right) \tag{1}$$

where $N$ is the total number of animals per unit (pen) and is composed of the number of animals in each stage ($n(S) + n(I_1) + n(I_2) + n(D) + n(C_1) + n(C_2)$). When an animal reaches the infectious state ($I_2$), it can infect other pigs in the stable. When the incubation period ($i$) has elapsed, the pig will start showing clinical symptoms and can therefore be detected ($D$). In the model, it is assumed that livestock owners will first detect symptomatic animals and then react according to the rules. Thus, if ASF is suspected, the official veterinarian must be contacted so that blood and/or tissue samples can be taken and sent to the appropriate reference laboratory for PCR testing. Once an animal reaches status $C_1$, the holding where it is located is officially classified as an infected holding, and all legally required safety measures are implemented. The culled animals must be taken to an RP to dispose of them harmlessly. In Figure 2, $r$ stands for the required time until the culled animals are removed

from the holding. In the best case, this is completed immediately. However, if there are bottlenecks in the RPs, $r$ would increase, meaning animals would have to remain on the holding until they can be carried away.

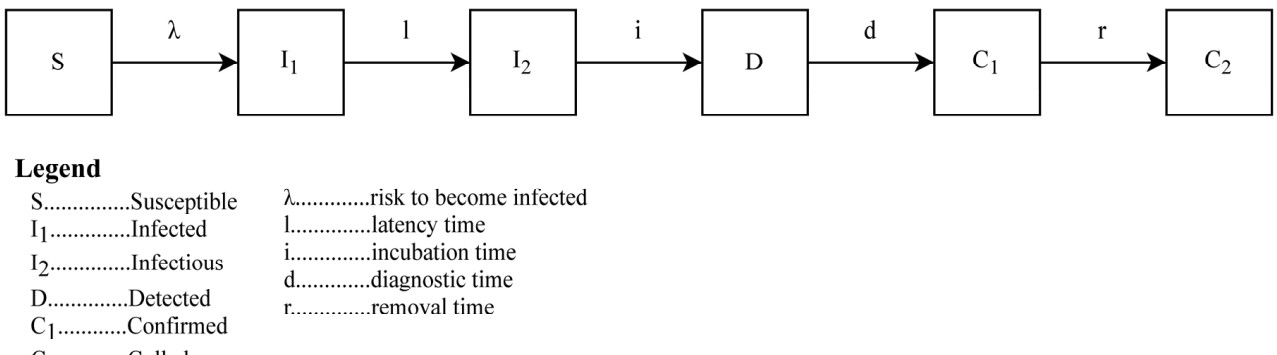

**Legend**

| | | | |
|---|---|---|---|
| S...............Susceptible | | $\lambda$..............risk to become infected | |
| $I_1$..............Infected | | l..............latency time | |
| $I_2$..............Infectious | | i..............incubation time | |
| D..............Detected | | d.............diagnostic time | |
| $C_1$............Confirmed | | r..............removal time | |
| $C_2$............Culled | | | |

**Figure 2.** $SI_1I_2DC_1C_2$ model.

**Table 2.** Model parameters and their values selected as defaults in the simulation.

| Component | Value | Unit | Description | Source |
|---|---|---|---|---|
| latency time [l] | 4 | Day | Period from infection to onset of infectivity | Guinat et al. [6] Guinat et al. [15] Pietschmann et al. [61] |
| incubation time [i] | 15 | Day | Period from infection to onset of symptoms | Austrian Agency for Health and Food Safety [70] |
| diagnostic time [d] | 24 | Hour | Period from onset of symptoms to receiving laboratory result | Information from reference laboratory |
| transmission rate within holding [β] | 0.3 | - | Number of secondary infections originating from an infectious entity per time unit | Guinat et al. [6] Eblé et al. [71] |
| radius protection zone | 3 | Kilometre | Certain disease eradication measures in this zone come into force | ASF-Regulation 2005 [68] Regulation (EU) 2016/429 [67] |
| radius surveillance zone | 10 | Kilometre | Certain disease eradication measures in this zone come into force | ASF-Regulation 2005 [68] Regulation (EU) 2016/429 [67] |
| radius infection zone | 40 | Kilometre | Zone in which the disease can be spread | Assumption by the authors |
| transmission rate infection zone | 0–10 | ‰ | See description of β | Assumption by the authors |
| transmission rate infection zone after confirmation | 0–10 | ‰ | See description of β | Assumption by the authors |
| initial farm type | 1–4 | - | Farm type of first infected pig holding (1 breeding farm, 2 rearing farm, 3 fattening farm, 4 combined farm) | Austrian Swine Health Regulation 2016 [72] |
| federal state | 1–4 | - | Federal state of first outbreak (1 Lower Austria, 2 Upper Austria, 3 Styria, 4 Carinthia) | - |
| outbreak start time | 987 | Day | Time when first pig becomes infected | Assumption by the authors |
| outbreak duration | 365 | Day | The transmission rate remains at the set level for this duration. Afterwards it is set to zero | Assumption by the authors |

Table 2 presents the model parameters and the values used. A range of values is usually used for epidemiological parameters (latency time, etc.) since a precise determination cannot be made. Nonetheless, herein a fixed value was used to make the test scenarios comparable. Based on different assumptions about the time–space context in which wild boars can spread ASF, a fixed radius of 40 km around the first infected farm was defined as the infection zone. Due to the increased number of infected wild boars in this zone, an increased risk of infection for pig holdings was assumed.

The spread between pig holdings was simulated as a standard SEIR model and modelled by a statechart (Figure 3). Transitions between states can be made in different ways, two of them are shown in Figure 3 using the icons in the arrows. The letterhead represents a specific message that triggers the transition. For example, such a message can be triggered automatically after infection and cause the transition from "Exposed" to "Infectious". The clock symbol represents a duration that triggers the transition after expiration.

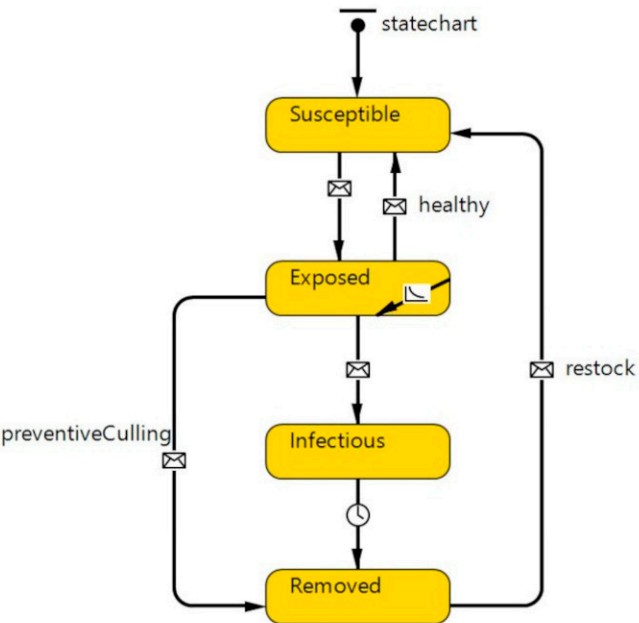

**Figure 3.** Statechart of the SEIR model on an individual pig holding level.

The exposed status is given to all holdings located in the infection zone. Therefore, these exposed holdings have a higher daily probability of being infected via wild boars or other contaminated material (see Table 2: transmission rate infection zone). This probability is difficult to determine and represents a parameter that the simulation user can choose. Contact holdings, i.e., farms that have received animals from an infected holding or have delivered animals to such a holding in the last 15 days, must also be tested for the presence of the virus and have the status exposed as well. If such a contact holding tests positive for ASF, a protection and surveillance zone is established around it. However, these holdings can be located in or outside the infection zone. If they are outside, the infection pressure does not automatically increase in the surrounding area (the transmission is assumed here to be via direct contact and not via wild boars). At an infectious holding, the animals are culled, and it obtains the status removed until the time prescribed by law has elapsed, and the holding can be restocked under specific legal requirements. When an exposed holding remains uninfected (healthy) until the end of the outbreak duration, it changes back to the status susceptible.

## 4. Numerical Studies

A total of 16,344 entities consisting of 40% fattening farms, 37% combined farms, 18% rearing farms, 4% breeding farms, 18 slaughterhouses, 24 assembly centres, and six RPs represent the pork supply chain in four selected federal states (Lower Austria, Upper

Austria, Styria, and Carinthia) of Austria totalling 57,098 km². These federal states cover 96% of the total pig population and 84% of the pig farms in Austria; 1027 municipalities were considered. Three RPs can process categories one and two of animal material (Figure 4). According to EU Regulation (EC) 1069/2009, this mainly includes specified risk material [73]; therefore, these RPs are responsible for the recovery of carcasses in the case of ASF. Our data basis was provided by the Federal Ministry of Social Affairs, Health, Care and Consumer Protection (BMSGPK) via the statistical authority Statistics Austria. These data were submitted anonymously to ensure that no inferences about individual farms could be derived. All farms in a municipality were anchored in the municipality centre. The spatial distribution of the entities is shown in Figure 4 as well as the federal states and their number of farms and stocking density.

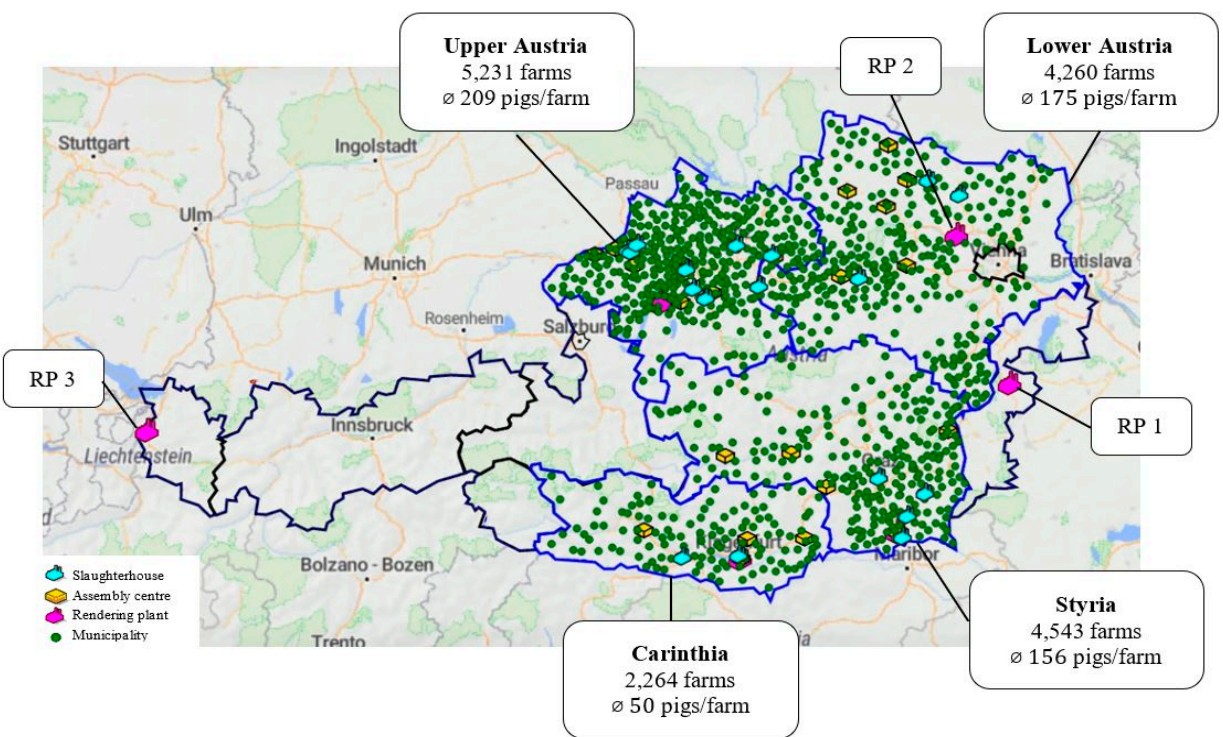

**Figure 4.** Spatial distribution and figures of domestic pig farms and rendering plants (RP) in selected federal states of Austria.

In preliminary tests, we used 120 scenarios to evaluate the effects of the first infected farm type and municipality as well as the outbreak start time. These tests showed significant differences in the number of infected farms and culled pigs depending on where (farm type and municipality) ASF first appeared. However, the outbreak start time did not show substantial differences. Therefore, 576 scenarios were defined to answer the research question. In each federal state, three municipalities were selected based on their number of farms and pig stock. These two criteria resulted in high, medium, and low relevance categories. Each municipality had 48 different parameter combinations. The parameters that were changed are shown in Table 3, including initial farm type, i.e., the farm type of the first infected farm (four options), the transmission rate (six options), and the control strategy (two options). With SC, animals suspected of being infected with ASF can be culled even before an official laboratory result is available, which is the main difference between the two assessed control strategies. In the simulation, due to references from corresponding laboratories in Austria, it was estimated that it would take 24 h to take samples from pigs on the suspected holding, send them to and evaluate them in the laboratory, and announce the result. During the laboratory analysis time, the management measures according to

Regulation (EU) 2016/429 can already be established using SC [67]. Therefore, contact holdings can be identified earlier to interrupt the transmission chain.

**Table 3.** Overview of the numerical studies and the included parameter variations.

| Initial Farm Type | Federal State | Municipality | | | Transmission Rate | | | Control Strategy |
|---|---|---|---|---|---|---|---|---|
| | | Abbr. | Number of Farms | Number of Pigs | Nr. | Infection Zone | Infection Zone after Confirmation | |
| 1 2 3 4 | Lower Austria | A | 56 | 47,527 | 1 2 | 1‰ 2‰ | 0.1‰ 0.2‰ | SS SC |
| | | B | 15 | 2755 | 3 4 | 3‰ 4‰ | 0.3‰ 0.4‰ | |
| | | C | 15 | 963 | 5 6 | 5‰ 1% | 0.5‰ 1‰ | |
| 1 2 3 4 | Upper Austria | D | 70 | 33,573 | 1 2 | 1‰ 2‰ | 0.1‰ 0.2‰ | SS SC |
| | | E | 38 | 18,419 | 3 4 | 3‰ 4‰ | 0.3‰ 0.4‰ | |
| | | F | 12 | 570 | 5 6 | 5‰ 1% | 0.5‰ 1‰ | |
| 1 2 3 4 | Styria | G | 139 | 48,249 | 1 2 | 1‰ 2‰ | 0.1‰ 0.2‰ | SS SC |
| | | H | 38 | 4233 | 3 4 | 3‰ 4‰ | 0.3‰ 0.4‰ | |
| | | I | 9 | 412 | 5 6 | 5‰ 1% | 0.5‰ 1‰ | |
| 1 2 3 4 | Carinthia | J | 32 | 10,185 | 1 2 | 1‰ 2‰ | 0.1‰ 0.2‰ | SS SC |
| | | K | 30 | 687 | 3 4 | 3‰ 4‰ | 0.3‰ 0.4‰ | |
| | | L | 11 | 100 | 5 6 | 5‰ 1% | 0.5‰ 1‰ | |

The simulation was performed on a computer with 60 GB RAM with an Intel® Core™ i7-3930K CPU, 3.20 GHz and Windows 10 as the operating system. The runtime of all scenarios together led to a total computing time of 387 h.

## 5. Results

The scenarios were evaluated based on their impact on the KPIs. The initial farm type did not show any significant difference in the data, so the average value over the four farm types was used as a basis in the following table. Table 4 shows the results per municipality (Mun.) divided into SS and SC. The mean value over all transmission rates is given by μ.

All three municipalities in Upper Austria (*D*, *E*, *F*) had the highest number of infected farms and emergency slaughtered animals compared to the municipalities of the other federal states. The concentration of the three most affected municipalities in Upper Austria could be related to the high pig population density in this area. The second most affected municipalities were in Styria, especially *G* and *H*, as Styria belongs to a densely populated area of pig farms and pigs too. However, municipality *I* in Styria reflected a different pattern: due to its geographic location close to the Alps, surrounded by few farms, one of the lowest dispersion scenarios took place here. Municipalities *A*, *B*, and *C* had high numbers of culled pigs, too, as Lower Austria has a high density of pigs. Municipality *B* is located on the border of Upper Austria and therefore had the highest number of infected farms and culled pigs in Lower Austria. Municipality *A* had the highest number

of farms and pigs compared to the other Lower Austrian municipalities but still had a lower number of infected farms and animals than municipality *C* because it is located in a sparsely populated area. Therefore, the geographic location of the farms can be concluded as a relevant characteristic for the overall intensity of the outbreak. The transmission rate further influenced the outbreak intensity. With the increasing transmission rate, more farms were infected. However, this trend was not continuous in the number of culled pigs. Higher transmission rates could also lead to lower culls because the farm sizes vary considerably.

**Table 4.** Overview of the numerical output per municipality.

| | **SS** | | | | | | | | | | | | |
| --- | --- | --- | --- | --- | --- | --- | --- | --- | --- | --- | --- | --- | --- |
| **Mun.** | **Number of Infected Farms** | | | | | | | **Number of Culled Pigs** | | | | | |
| | **Transmission Rate** | | | | | | **μ** | **Transmission Rate** | | | | | | **μ** |
| | **1** | **2** | **3** | **4** | **5** | **6** | | **1** | **2** | **3** | **4** | **5** | **6** | |
| A | 32 | 74 | 103 | 130 | 164 | 277 | **130** | 7472 | 34,289 | 50,510 | 45,272 | 31,088 | 72,210 | **40,140** |
| B | 102 | 183 | 274 | 387 | 472 | 813 | **372** | 11,576 | 19,286 | 40,517 | 43,152 | 56,213 | 97,674 | **44,736** |
| C | 39 | 73 | 110 | 140 | 174 | 296 | **139** | 18,838 | 27,026 | 44,486 | 35,627 | 50,903 | 87,134 | **44,002** |
| D | 104 | 215 | 324 | 416 | 503 | 920 | **414** | 30,110 | 79,480 | 118,456 | 137,306 | 177,721 | 320,340 | **143,902** |
| E | 97 | 205 | 300 | 384 | 497 | 888 | **395** | 27,435 | 55,379 | 84,161 | 113,867 | 146,631 | 269,025 | **116,083** |
| F | 98 | 203 | 324 | 412 | 524 | 940 | **417** | 48,344 | 37,871 | 50,981 | 89,482 | 103,567 | 111,461 | **88,126** |
| G | 74 | 145 | 219 | 282 | 351 | 635 | **284** | 18,357 | 38,882 | 67,278 | 87,807 | 92,622 | 177,093 | **80,340** |
| H | 73 | 136 | 204 | 267 | 313 | 595 | **264** | 12,564 | 20,162 | 32,755 | 40,761 | 55,489 | 97,957 | **43,281** |
| I | 23 | 44 | 60 | 84 | 94 | 171 | **79** | 505 | 1581 | 1889 | 3591 | 2700 | 6232 | **2749** |
| J | 48 | 95 | 148 | 198 | 229 | 416 | **189** | 5364 | 9868 | 13,066 | 18,124 | 20,476 | 33,230 | **16,688** |
| K | 43 | 84 | 114 | 152 | 204 | 359 | **159** | 1793 | 2642 | 4838 | 5490 | 9526 | 13,076 | **6227** |
| L | 34 | 63 | 92 | 130 | 155 | 277 | **125** | 431 | 990 | 1746 | 2491 | 2681 | 5076 | **2202** |
| | **SC** | | | | | | | | | | | | |
| **Mun.** | **Number of Infected Farms** | | | | | | | **Number of Culled Pigs** | | | | | |
| | **Transmission Rate** | | | | | | **μ** | **Transmission Rate** | | | | | | **μ** |
| | **1** | **2** | **3** | **4** | **5** | **6** | | **1** | **2** | **3** | **4** | **5** | **6** | |
| A | 33 | 66 | 96 | 121 | 158 | 274 | **125** | 7992 | 16,329 | 29,044 | 33,163 | 36,071 | 75,078 | **32,946** |
| B | 100 | 177 | 269 | 375 | 451 | 839 | **369** | 7427 | 34,289 | 50,510 | 45,272 | 31,088 | 72,210 | **40,140** |
| C | 34 | 72 | 99 | 131 | 164 | 302 | **134** | 12,313 | 17,605 | 29,344 | 32,006 | 41,700 | 98,802 | **38,628** |
| D | 102 | 203 | 315 | 427 | 501 | 917 | **411** | 34,009 | 65,124 | 99,049 | 150,449 | 163,148 | 314,547 | **137,721** |
| E | 90 | 190 | 286 | 401 | 466 | 900 | **389** | 35,850 | 58,677 | 79,542 | 112,113 | 129,433 | 265,302 | **113,486** |
| F | 100 | 208 | 319 | 417 | 519 | 917 | **413** | 15,722 | 38,449 | 54,055 | 86,934 | 104,556 | 188,144 | **81,310** |
| G | 86 | 146 | 210 | 287 | 347 | 624 | **283** | 20,342 | 41,969 | 57,829 | 83,659 | 94,812 | 168,389 | **77,833** |
| H | 63 | 134 | 206 | 257 | 331 | 580 | **262** | 13,174 | 23,934 | 37,351 | 41,578 | 52,544 | 97,330 | **44,318** |
| I | 22 | 38 | 56 | 80 | 96 | 170 | **77** | 757 | 1287 | 2734 | 2572 | 3984 | 7528 | **3144** |
| J | 54 | 92 | 146 | 186 | 220 | 429 | **188** | 5799 | 9005 | 12,562 | 16,413 | 19,112 | 35,754 | **16,441** |
| K | 42 | 84 | 115 | 159 | 175 | 356 | **155** | 1817 | 2661 | 4774 | 5935 | 6612 | 15,347 | **6024** |
| L | 29 | 69 | 92 | 132 | 149 | 265 | **123** | 516 | 1072 | 2334 | 2485 | 4270 | 5400 | **2679** |

Another simulation output concerned the number of days on which RPs were overutilized. The capacities of the RPs were set based on the available capacity per day, calculated from the annual amount of processed animal material provided on the respective RP's website. However, in our scenario, 100% of the available capacity was set aside for utilizing pigs, thus excluding the processing of other materials. Table 5 shows the number of days the respective RP had no more capacity available for each municipality, transmission rate and control strategy. The number of days resulted from the maximum value of the four farm types as the initially infected farm. When all three RPs reached their capacity limit, waiting times for the pig holdings concerning the collection of culled animals occurred. Due to the long distance, RP 3 was only served when RP 1 and 2 had no more available capacity. Thus, the holdings will have delayed pickups as soon as all three RPs within one municipality have a value higher than zero in Table 5. Overall, this occurred equally often with both strategies. However, when comparing the total number of days at which each RP was overutilized per transmission rate, the SS had more days that were overutilized. On the one hand, the data presented in Table 5 again show the focus on municipalities *D*, *E*, and *F* in Upper Austria, which had the highest number of days where RPs were overutilized. On the other hand, as infection rates increased, an increase in the overutilized days was seen.

**Table 5.** Number of days with exhausted capacity in rendering plants.

| Transmission Rate | 1 | | | | | | 2 | | | | | | 3 | | | | | |
|---|---|---|---|---|---|---|---|---|---|---|---|---|---|---|---|---|---|---|
| Strategy | SS | | | SC | | | SS | | | SC | | | SS | | | SC | | |
| | **Number of Days When RP Is Overutilized** | | | | | | | | | | | | | | | | | |
| | Rendering Plant | | | | | | Rendering Plant | | | | | | Rendering Plant | | | | | |
| Municipality | 1 | 2 | 3 | 1 | 2 | 3 | 1 | 2 | 3 | 1 | 2 | 3 | 1 | 2 | 3 | 1 | 2 | 3 |
| A | 1 | 2 | 0 | 1 | 2 | 0 | 12 | 13 | 7 | 3 | 4 | 2 | 8 | 11 | 2 | 5 | 6 | 1 |
| B | 1 | 2 | 0 | 1 | 2 | 1 | 1 | 1 | 1 | 1 | 3 | 0 | 2 | 5 | 1 | 1 | 4 | 1 |
| C | 7 | 12 | 6 | 1 | 4 | 0 | 6 | 6 | 4 | 1 | 6 | 0 | 7 | 11 | 4 | 4 | 8 | 0 |
| D | 2 | 4 | 0 | 2 | 4 | 1 | 3 | 13 | 0 | 4 | 13 | 2 | 9 | 16 | 3 | 9 | 13 | 2 |
| E | 1 | 4 | 0 | 7 | 9 | 6 | 2 | 11 | 1 | 12 | 18 | 9 | 2 | 11 | 1 | 4 | 11 | 2 |
| F | 5 | 12 | 2 | 0 | 3 | 0 | 1 | 11 | 1 | 1 | 6 | 0 | 5 | 12 | 3 | 3 | 7 | 1 |
| G | 2 | 1 | 0 | 2 | 2 | 1 | 4 | 1 | 1 | 6 | 4 | 3 | 8 | 6 | 0 | 8 | 6 | 1 |
| H | 2 | 2 | 1 | 3 | 2 | 1 | 1 | 1 | 0 | 2 | 1 | 0 | 3 | 2 | 1 | 6 | 4 | 1 |
| I | 0 | 0 | 0 | 0 | 0 | 0 | 0 | 0 | 0 | 0 | 0 | 0 | 0 | 0 | 0 | 0 | 0 | 0 |
| J | 2 | 1 | 0 | 2 | 1 | 0 | 2 | 1 | 1 | 1 | 1 | 0 | 1 | 1 | 0 | 1 | 1 | 0 |
| K | 0 | 0 | 0 | 0 | 0 | 0 | 0 | 0 | 0 | 0 | 0 | 0 | 1 | 0 | 0 | 0 | 0 | 0 |
| L | 0 | 0 | 0 | 0 | 0 | 0 | 0 | 0 | 0 | 0 | 0 | 0 | 0 | 0 | 0 | 0 | 0 | 0 |

| Transmission Rate | 4 | | | | | | 5 | | | | | | 6 | | | | | |
|---|---|---|---|---|---|---|---|---|---|---|---|---|---|---|---|---|---|---|
| Strategy | SS | | | SC | | | SS | | | SC | | | SS | | | SC | | |
| | **Number of Days When RP Is Overutilized** | | | | | | | | | | | | | | | | | |
| | Rendering Plant | | | | | | Rendering Plant | | | | | | Rendering Plant | | | | | |
| Municipality | 1 | 2 | 3 | 1 | 2 | 3 | 1 | 2 | 3 | 1 | 2 | 3 | 1 | 2 | 3 | 1 | 2 | 3 |
| A | 16 | 21 | 9 | 5 | 9 | 5 | 2 | 5 | 1 | 4 | 5 | 0 | 7 | 15 | 5 | 6 | 12 | 5 |
| B | 2 | 4 | 0 | 0 | 2 | 0 | 1 | 5 | 0 | 1 | 4 | 1 | 3 | 13 | 3 | 1 | 14 | 0 |
| C | 3 | 4 | 2 | 2 | 6 | 0 | 9 | 18 | 2 | 5 | 16 | 3 | 6 | 27 | 2 | 6 | 24 | 5 |
| D | 6 | 25 | 1 | 7 | 27 | 3 | 8 | 30 | 2 | 9 | 28 | 3 | 20 | 64 | 8 | 17 | 54 | 8 |
| E | 2 | 19 | 0 | 5 | 14 | 1 | 9 | 26 | 1 | 9 | 21 | 3 | 14 | 55 | 3 | 9 | 54 | 2 |
| F | 4 | 13 | 2 | 4 | 12 | 3 | 3 | 15 | 0 | 3 | 13 | 1 | 7 | 38 | 1 | 5 | 33 | 1 |
| G | 13 | 8 | 5 | 11 | 5 | 1 | 9 | 4 | 0 | 12 | 4 | 1 | 25 | 9 | 2 | 26 | 8 | 2 |
| H | 4 | 2 | 0 | 6 | 1 | 1 | 5 | 3 | 1 | 6 | 3 | 0 | 9 | 3 | 0 | 10 | 3 | 1 |
| I | 0 | 0 | 0 | 0 | 0 | 0 | 0 | 0 | 0 | 0 | 0 | 0 | 0 | 0 | 0 | 0 | 0 | 0 |
| J | 2 | 1 | 1 | 1 | 1 | 0 | 1 | 1 | 0 | 1 | 1 | 0 | 1 | 1 | 0 | 2 | 0 | 0 |
| K | 1 | 0 | 0 | 0 | 0 | 0 | 1 | 0 | 0 | 0 | 0 | 0 | 0 | 0 | 0 | 1 | 1 | 0 |
| L | 0 | 0 | 0 | 1 | 1 | 0 | 0 | 0 | 0 | 0 | 0 | 0 | 0 | 0 | 0 | 0 | 0 | 0 |



The results also allowed a comparison of the two control strategies. On average, SC led to three fewer infected farms and 2699 fewer emergency slaughtered animals per municipality. However, a minimum detection time of 24 h was chosen, which may also range up to 48 h, according to the experts' statement. With an increasing delay in detection, increasing differences in the strategies can be assumed, highlighting SC as an important measure for disease control. Nevertheless, even with the small-time window of 24 h, a reduction in infected farms and emergency slaughtered pigs is possible.

## 6. Discussion

The presented simulation model can represent different ASF outbreak scenarios and depicts the impacts on the primary production of pork in Austria in the context of various influencing factors and control strategies. In addition, assumptions made in the simulation, such as the infection radius and transmission rate, can be manipulated by the user. In this way, the model can act as a DSS that can be used to train decision makers. This DSS has the capability to test different outbreak locations, transmission scenarios, and more to prepare for various crisis scenarios and to strengthen the resilience of the pork supply chain. Although the focus of this simulation study was not to develop an epidemiological model to calculate the infection rate, reproduction number, or similar, previously developed models and indicators were taken for granted to develop an individual $SI_1I_2DC_1C_2$ model. Such individual-based models with a large number of agents (pigs) are computationally intensive, which is why currently available models aggregate individuals and focus on domestic pigs or wild boars within a limited area [51]. The grouping of several pigs to one agent has contributed to a significant reduction in agents, although the allocation of these groups is possible and is performed automatically by the developed algorithm. At the same time, however, each farm was represented individually at the municipality level, which shows a very high level of detail. Besides, this model does not include international commodity flows since exports and imports would be significantly restricted during an ASF outbreak. The transport of culled animals for destruction abroad represents an enormous risk and is therefore not an option from an epidemiological point of view. Therefore, the essential task of resilience research is to maintain the national disposal chain and prevent any delays and thus the spread of the virus. This model has been crucial in identifying the bottlenecks in this disposal chain under different assumptions and scenarios.

It must be noted that the quality of the simulation depends on the available input data. Some data are subject to the General Data Protection Regulation or are not published or disclosed by the companies. The validity of the results falls with the availability of this data. Some resources, such as official veterinarians, transportation, and laboratory capacity, were assumed to be infinite, as experts did not consider them limiting. However, within a large-scale ASF crisis event, these resources could be overutilized and become a bottleneck resource too. This possibility could be evaluated in further steps when these capacities are known. Nevertheless, in this simulation, we had the opportunity to cooperate with the BMSGPK and obtain valuable real data on pig holdings. Gaps in the data sets, such as the supply relationships, could be compensated through the developed algorithms. Therefore, this model can simulate realistic outbreaks of ASF in Austria. Due to the possibility of adjusting the simulation's input individually to the current state of knowledge, a wide variety of outbreak courses and options for action can be represented and used for decision makers. One possible action option for farmers could be precautionary measures in terms of biosecurity. The effects of such measures could be evaluated in further model calculations. The singularity of this model in Austria, but also internationally, is based primarily on the mentioned possibilities. Linking the simulation to the Austrian veterinary information system, for instance, would be one way of ensuring that the data are up to date and would represent a further improvement of the model. We defined three KPIs that could be positively affected with the appropriate control strategy. Accordingly, a particularly interesting finding of this study was that applying SC may reduce the overall impact of an

ASF outbreak. Nevertheless, the combination of SC and SS or other strategies beyond the scope established by law could be tested in further research.

**Author Contributions:** Conceptualization, Y.K., P.H., J.B. and K.J.D.; methodology, Y.K., P.H. and C.F.; software, Y.K., C.F. and P.H.; validation, Y.K. and P.H.; formal analysis, Y.K.; investigation, Y.K., P.H., C.F., J.B., K.J.D., M.S. and R.F.; resources, Y.K. and P.H.; data curation, Y.K., P.H., M.S. and R.F.; writing—original draft preparation, Y.K. and P.H.; writing—review and editing, Y.K., P.H., C.F., J.B., M.S., R.F. and K.J.D.; visualization, Y.K.; supervision, P.H.; project administration, Y.K. and P.H.; funding acquisition, C.F., K.J.D., J.B. and P.H. All authors have read and agreed to the published version of the manuscript.

**Funding:** This work was written within the research project NutriSafe supported by the Federal Ministry Republic of Austria for Agriculture, Regions, and Tourism within the security research funding programme KIRAS (project number: 867015) and the German Federal Ministry of Education and Research (project number: 13N15070-13N15076) within the civil security research programme.

**Institutional Review Board Statement:** Not applicable.

**Informed Consent Statement:** Not applicable.

**Data Availability Statement:** Not applicable.

**Acknowledgments:** We would like to thank Melinda Bozic from the Austrian Federal Ministry of Social Affairs, Health, Care, and Consumer Protection for providing the relevant data basis. Further, we would like to thank Michael Kugler from the Austrian Federal Chancellery, who provided valuable input for the research focus from a governmental perspective. We also thank our colleague Klaus-Dieter Rest, who contributed important inputs to the data preparation.

**Conflicts of Interest:** The authors declare no conflict of interest.

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
