# Peer review of "Facilitating Resilience during an African Swine Fever Outbreak in the Austrian Pork Supply Chain through Hybrid Simulation Modelling"

_agriculture, doi:10.3390/agriculture12030352_

Round 1

Reviewer 1 Report

Line 168 - the abbreviation "RP" should be clarified.

Line 176 - EU Directive on the ASF regulation (2002/60) was repealed on April 2021 by Regulation 2016/429 and the control measures are included in the delegated Regulation 2020/687 and Regulation 2020/689. Lines 176-181 should be corrected accordingly and in the rest of the text WU Directive should be changed to the EU Regulation.

Lines 195-197 - according to the Delegated Regulation 2020/687, the minimum period duration of measures in the protection zone for ASF is 15 days and in the surveillance zone - 30 days. It seems, that the difference in the duration of the measures may influence the overall calculation?

Line 212-217 - this is the truly theoretical approach. The current epidemic with the genotype II ASF virus strain in the domestic pig holding starts without any symptoms at the earliest stage and sudden death of pig is the only symptom, that wise the presence of ASF on the farm may remain undetected for the several weeks. The clinical symptoms in the stable appear not at the earliest stage, but when the disease has been progressed and spread within a farm, affecting many animals. In most cases, pigs will die on 3-7 days post-infection.

Line 229 - the speed of natural propagation of ASF in wild boar populations between 2.9 and 11.7 km/year (the median velocity of infection in Belgium, Estonia, Hungary, Latvia, Lithuania and Poland (EFSA, 2020). It is recommended in case of infection in the wild boar population to establish the infected area at 200 sq km, which have a radius of 8 km. The currently proposed 40 km fixed radius will cover approx. 5000 sq km, which is not realistic at all. The recalculation might be necessary to have the scenario close to the realistic one.

Incubation time according to the OIE: Incubation period in nature is usually 4–19 days; acute form 3–4 days. For the purposes of the OIE
Terrestrial Animal Health Code, the incubation period in Sus scrofa shall be 15 days.  https://www.oie.int/app/uploads/2021/03/oie-african-swine-fever-technical-disease-card.pdf

It should be noted as well, that ASF is not an airborne disease and the spread did not occur without direct or indirect contact, which influence much on the transmission rate. The proposed calculation and the model itself would fit more to FMD, HPAI or CSF.

As an additional risk parameter the biosecurity of pig holdings in the calculated area can be included, while it has been proven by the ASF infected countries, that even if the infected area is established due to ASF in the wild boars, biosecure farms can remain not affected despite the virus circulation in the proximity.

Reviewer 2 Report

The manuscript is well and clearly written, without any methodological or linguistic mistakes.

Presented topic is an interesting study for the agri-food sector as well as for the breeders and scientist workin with the African Swine Fever in practice.

The only doubts are related to the Fig. 1., and I strongly recommend to correct it or re-drawn, making more clear for the reader.

In my opinion, the article should have a small correct related to mentioned aspect.  

Round 2

Reviewer 1 Report

The current version is improved and unclarities were removed/explained.